# The Brewing Industry and the Opportunities for Real-Time Quality Analysis Using Infrared Spectroscopy

**Glen Fox** [1,2]

1    Department of Food Science & Technology, University of California, Davis, CA 95616, USA; gpfox@ucdavis.edu
2    Centre for Nutrition and Food Science, Queensland Alliance for Agriculture & Food Innovation, University of Queensland, St Lucia, QLD 4072, Australia

**Abstract:** Brewing is an ancient process which started in the middle east over 10,000 years ago. The style of beer varies across the globe but modern brewing is very much the same regardless of the style. While there are thousands of compounds in beer, current methods of analysis rely mostly on the content of only several important processing parameters such as gravity, bitterness, or alcohol. Near infrared and mid infrared spectroscopy offer opportunities to predict dozens to hundreds of compounds simultaneously at different stages of the brewing process. Importantly, this is an opportunity to move deeper into quality through measuring wort and beer composition, rather than just content. This includes measuring individual sugars and amino acids prior to fermentation, rather than total °Plato or free amino acids content. Portable devices and in-line probes, coupled with more complex algorithms can provide real time measurements, allowing brewers more control of the process, resulting in more consistent quality, reduced production costs and greater confidence for the future.

**Keywords:** barley; malt; beer; near infrared; mid infrared; fermentation; alcohol; in-line quality

## 1. Introduction

Brewing is one of man-kind's oldest food processes. The earliest time fermented beverages were produced was many thousands of years ago. The early fermented beverages were produced from an uncontrolled process due to variable materials used, mainly based on seasonal grains, herbs and fruits and more importantly, the region and season the brew was made. The relevance of the growing region or season effects impact on the different strain of yeast and other organisms present which would give rise to spontaneous fermentations. The alcoholic beverage produced was probably safer than drinking water.

Jumping ahead a few thousand years, and we live in an age of highly technical processing, where consistent quality is required. Costs and balance sheets drive the malting and brewing businesses which pressures companies to innovate to reduce costs and remain competitive. One area where breweries especially can monitor quality is through using real-time sensors. The malting and brewing industries have adopted basic real-time sensing such as temperature, pH or $O_2$ at various stages of the processes. However, there is scope for further utilization of sensor technology, more specifically in the brewing process [1].

This review will discuss where infrared (both near infrared (NIR) and mid-infrared (MIR)) have been used to screen for grain quality, malt quality, hop quality and within the brewing process and provide insights to where additional in-line technology would be advantageous to the industry,

particularly in the area of composition. The major components of barley particularly will get the most attention, i.e., protein, starch and non-starch polysaccharides as these have been by far the most studied in terms of malt and beer quality, but also receiving the most attention by infrared practitioners.

## 1.1. Background on Infrared (Near and Mid)

Near infrared and mid infrared are two adjacent spectral regions, with NIR being immediately after the visible, moving to the longer wavelength region [2] (Figure 1). When a sample matrix is presented to the infrared light, molecular vibrations are measured through the absorbances at each wavelength and from this absorbances associated to specific chemical bonds are made. Strongly absorbing bonds are C–H, N–H, O–H, C–C and C=C where these make up most of the structures in most organic substances, especially, plant material (barley and hops) and liquids (wort and beer). Predictive models are built on samples that have had a reference analysis carried out as well as having the samples scanned on the infrared spectrophotometer with spectral data captured. The spectral data is modelled against the reference data using algorithms such as multiple linear regression (MLR), partial least squares (PLS) or more advanced machine learning such as artificial neutral networks (ANN). These algorithm programs can provide real-time data predictions if built into the instruments or attached computer. Alternatively, the spectra can be captured and results predicted off-line.

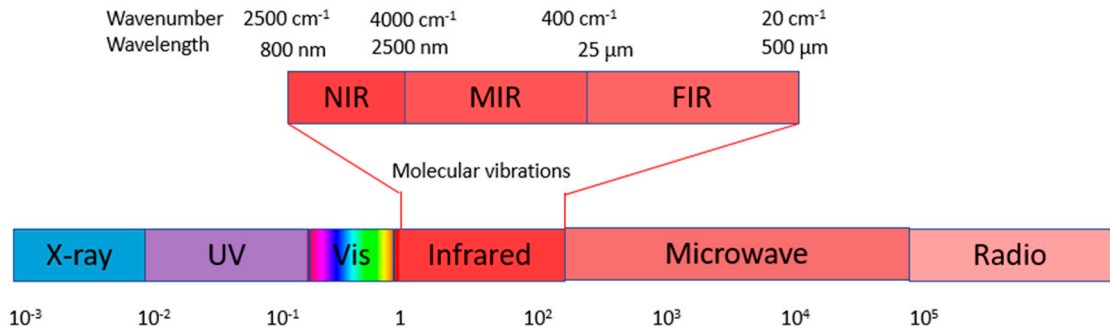

**Figure 1.** Spectral range for near infrared (NIR) and mid-infrared (MIR) showing wavelengths (nm) and wavenumbers (cm).

In-line probes and hand-held infrared devices now provide opportunities for additional data to be captured real-time which will provide brewers with more data to control the brewing process and final quality.

## 1.2. Cereal Breeding

Early forms of beer were very much different to modern beer styles but the base raw materials of grains, specifically barley (*Hordeum vulgare*), have been around for 1000 s of years and significant effects over the last century have focused on breeding and growing high quality malting barley. There has been much less effect to produce malting quality wheat, sorghum or rice, the other cereals used in beer production. To facilitate improving grain quality through breeding, primarily NIR and to a lesser extent MIR, have been used to screen for malting quality in barley breeding programs.

The selection of barley is based on physical grain characteristics such as large grain size and heavy grain weight, but also on protein content [3]. Lower protein content is preferred usually in the range of 9% to 13% (dry basis (db)). The low protein content should provide increased levels of starch and low levels of beta-glucan and arabinoxylan where each of these play a role in optimizing malt quality and then how efficient the brewing process will be. While these traits are selected on a content basis, the influence of each of these is based on the amount but more importantly the composition and this is where neither NIR nor MIR has been used within a breeding selecting context.

### 1.2.1. Protein

Malting barley is purchased from farmers and traded based on protein content and the most common way to rapidly measure content is using NIR. Until the last few years, when a farmer delivered a sample to a grain handling depot, that was the first time barley passed through an NIR. However, recent developments have seen NIR installed on grain harvesters for real time predictions of protein and moisture, coupled with yield data [4,5]. This allows farmers on broad-acre farms to have an understanding of the protein content within and between each paddock and potentially allow the farmer to exclude high protein grain accordingly. In addition, grain data can be related to nitrogen fertilizer inputs and the amount nitrogen extracted and stored in the grain.

Grain trading companies have networks of instruments across national programs to accumulate sufficient volumes for domestic and/or export markets. However, long before varieties are released, breeding programs use NIR to select for breeding lines within a protein range [6–9].

New barley varieties have been through years of testing in a breeding program. Intense selection is required to ensure potential new varieties can meet expectations of the farmers and the malting and brewing industries. Structured breeding trials grown over numerous locations and years provide grain for detailed quality testing where protein content is the first screen [10]. Its importance means that protein calibrations have been reported in numerous reports (see reviews: [11,12]). There are several **N**itrogen-**H**ydrogen (NH) absorbance regions and the use of spectra alone can be used for selection [8,13]. It has been shown that spectra can differ between growing regions and to a less extent underlying genetics [11,14]. Thus with a NIR spectrophotometer with a full spectra region it could be possible to do selections for protein and other traits without needing the support of any chemistry to develop calibrations. Spectral data was used to identify genetic traits in a breeding population where important genetic regions were identified [8,15]. Further, the use of NIR in assessing protein content in breeding, understanding the heritability of an important trait such as protein is important and it has been shown exactly that, i.e., heritability can be calculated for protein content using predicted protein data [16].

To fully understand the potential quality of breeding lines, and where protein is the key industry trait, other barley grain and malt quality traits have been correlated to protein content [6,7,17,18]. The success of any infrared studies is based on precise and accurate calibrations, where most research would expect an $R^2$ of >0.95 and Root Mean Standard Error Prediction < 0.5% for protein. The early instruments had a limited number of wavelengths and calibrations were built using multiple linear regression [19], but the latest instruments have 1000 s of wavelengths using PLS, ANN and other machine learning techniques [20].

Typical analysis of barley protein content is on bulk samples with 1000 s of grains in the sample tested. However, several studies have shown NIR has the capacity to scan and predict single grains and from that data show the distribution of protein content in a sample [21–23]. These studies showed variation between single kernels within a spike of barley as well as variation in other quality parameters based on grains sorted into sub-groups of protein content. The ideal protein level for malting barley is 10% to 11% (db) but current commercial crops vary in protein content within a paddock thus the variation is delivered in bulk for processing. For optimal malting barley, high through-put predictive single-kernel sorting would deliver this ideal protein range but limitations in speed of prediction, physical sorting and extra storage negate this at the commercial scale.

Protein content has been and will remain the first and key trait in any malting barley quality [24–27]. However, the content of protein is a summation of hundreds of individual proteins and other nitrogenous compounds in barley (and all other grains used in brewing). In producing malt and beer, hundreds of individual proteins and smaller peptides hydrolyzed from storage proteins remain through the brewing process, which includes a boiling stage, and go into finished beer [28]. Important individual proteins from barley and malt would be beta-amylase (from barley), alpha-amylase and limit dextrinase in malt but all three are inactivated during mashing [29]. However, if alpha-amylase remains active then the higher temperature of lautering will completely inactivate all enzymes [30]. However, some

proteins from barley that survive the malting and brewing process are lipid transfer protein (LTP), Protein Z and a number of inhibitors such as amylase/trypsin inhibitors (named chloroform methanol (CM) proteins) [28]. The challenges to build calibrations for individual proteins include the very low proportion being less than 1% of total protein and the modification of the protein in the purification process. However, it would be beneficial to the industry to have in-line assessment of key beer protein such as LTP due to its foam positive effect.

One of the largest individual proteins is hordein. It is the major storage protein in the endosperm and around 40% of total protein. Hordein is degraded during germination as a source of amino acids for the growing embryo, but in malting the amino acids provide a nitrogenous source for yeast during fermentation and larger hordein peptides can remain through into beer having a negative impact of foam proteins and potentially causing haze in beer [28]. Innovation NIR calibrations have been reported for hordein with the first using extracted hordein to identify key wavelengths in barley and malt and the second using the fourth derivative to build a PLS model [31,32]. The use of NIR to observe changes during the malting and mashing process would provide better indications of protein modification during germination and the amount of hordein peptides in beer.

Amino acids are assimilated by yeast during fermentation. Many proteins are partially hydrolyzed by enzymes such as endoproteinases [33,34] and carboxypeptidases [35,36] during the malting process and also the mashing process, depending upon the mashing temperature [37]. A measure of the free amino nitrogen, i.e., total amino acids present in wort is the standard method, with breweries requiring a minimum of 150 to 180 mg/mL to ensure full fermentation. Both NIR and FTIR have been used to identify amino acids in wort and beer [38]. Infrared sensors calibrated for amino acids and hordeins would be the most useful of the nitrogenous compounds that could be tracked during the brewing process as a way to observe potential yeast fermentation, and foam and haze stability, respectively.

### 1.2.2. Starch

Starch is the most abundant part of any cereal grain, making up around 65% of the total grain. The two starch polymers (amylopectin and amylose) make up the granules within the endosperm cells. The amylopectin and amylose differ by size and average number of glucose units making up the chains [39]. These starch granules are partially surrounded by hordein and during malting hordein is degraded by proteases to allow the starch degrading enzymes to hydrolyze the starch into smaller glucose sugars such as the disaccharide maltose, glucose–glucose. The natural biological relationship in grain development is a negative balance between starch and protein, and as such for the malting and brewing industries, a low protein high starch content is preferred. While starch is arguably one of the most important components from grain needed in brewing, the easier selection is for protein content. However, both NIR and MIR have been used to build calibrations for starch content and some starch properties [40–42]. Breeding programs have used calibrations for starch content and for genetic variants on amylopectin or amylose [13,40,43]. Further, other genetic mutations for traits such as high lysine or beta-glucan were genetically linked to changes in starch [13]. These mutations could be observed in spectra and separated using principle component analysis (PCA) [44] and other spectral analysis techniques [15].

Changes in composition from barley to malt have been reported using NIR [45] but changes in starch content had not been reported until very recently [46]. Malting is a controlled germination and drying process where starch is partially hydrolyzed and although most of the hydrolysis of starch occurs during mashing, the first stage of brewing NIR and MIR calibrations have been built for malt starch traits mostly as research or breeding selection calibrations. Considering the importance of starch in the brewing process, this would be an opportunity for the malting and brewing industry.

### 1.2.3. Non-Starch Polysaccharides—Beta-Glucan and Arabinoxylan

Another important barley trait that influences brewing quality is the non-starch polysaccharides which are made up of two polymers, beta-glucan and arabinoxylan. Like the two starch polymers,

both the amount and structure of beta-glucan and arabinoxylan is important. These two polymers are located in the cell walls of the endosperm and aleurone layers. Both of these can have negative effects in brewing if they are only partially hydrolyzed during malting and mashing, as they have the potential to increase wort viscosity [47,48] resulting in filtration problems [49]. The malting process reduces these to smaller oligosaccharides but they can still slow filtration in the brewery. Barley and malt beta-glucan NIR calibrations have been reported [8,40]. However, there are no reports for barley or malt arabinoxylan. The calibrations for beta-glucan save considerable time and costs as the beta-glucan reference method is complex. These calibrations are used by breeding programs and research groups whereas the barley grain industry and malt industries have not reported any applications. Ideally, the first commercial application would be for the malt industry as malt beta-glucan and arabinoxylan are correlated to the final beta-glucan in the brewers liquid (wort). In-line infrared sensors detecting the amount of both beta-glucan and arabinoxylan in the mash would give brewers the opportunity to monitor mashing and add glucanase or xylanase to hydrolyze the substrates further, reducing the risk of viscosity or filtration problems.

### 1.2.4. Minor Constituents

A number of other chemical components from the grains used in brewing including lipids, polyphenols, minerals and vitamins have been subject to infrared analysis either in the grain or perhaps in the malt [1,50–54]. These are contributors to beer in either positive or negatives ways with fatty acids being involved in yeast metabolism and ester production, but also lipid content having a negative impact on foam quality. Polyphenols are important in grain quality, but certain polyphenols can negatively influence beer stability by being responsible for haze. However, phenolic acids are important for some beer flavors and provide antioxidant capacity. By not measuring these the brewers will be missing important compounds that influence beer quality, whereas if brewers do measure these, even if only used as a potential way to trouble shoot then there are considerable benefits.

### 1.3. Hops

Another step, in the stabilization, flavor and aroma of beer, was the addition of hops (*Humulus lupulus*) in around the ninth century [55]. Hops were used in food to add flavor (bitterness) but more importantly, hops provided increased food preservation. Two major compounds from hops, alpha-acids and beta-acids, provide the bitterness in beer while essential oils give a range of aromas [55]. Hops can be added at different times of the brewing process resulting in different effects. For bitterness and aroma, hops either as the dried hop flowers or in a pellet form, are added during the first stage of boiling where the alpha-acids are isomerized and become more bitter. However, the vigorous boiling process will steam off some of the aromas, hence more aromatic hop varieties can be added at the end of boiling (called late hopping). The final stage where hops may be added is during fermentation where again addition aroma can be added. Fermentation is at much lower temperatures (usually below 20 °C) so no additional bitterness is added to the beer [30].

Over many decades, hop breeders have targeted individual compounds to give hop varieties a unique level of bitterness or enhanced aromas but there has been very limited application of infrared analysis of hop quality. NIR has been used for predicting bitter compounds (alpha-acids and beta-acids) in hops [56] while NIR and FTIR has been used to detect bittering units in beer [1,57]. Both NIR and MIR spectroscopy was used to successfully discriminate between hops varieties [58].

It is somewhat surprising that the use of infrared in hops has not received as much attention as the grains used in brewing. Measuring bitterness units is not a straightforward method but infrared calibrated against a liquid chromatography (LC) standard method, is a more rapid and potentially more accurate method, and could be measured in-line after boiling and the bitterness level has stabilized.

*1.4. Production of Wort then Beer*

1.4.1. Mashing

Wort is the liquid extracted during the mashing process (Figure 2) [59]. Milled malt is mixed with hot water over approximately one hour to extract water soluble components. To achieve this, enzymes synthesized during the malting process hydrolyze malt protein, starch, and non-starch polysaccharides as described above. During mashing, the typical sensors would be for temperature and pH. However, the addition of sensors to monitor the progression and level of the hydrolyzed products would give brewers an indication of the sugars and amino acids required for optimal fermentation, as well as potentially monitoring the amount of hydrolyzed hordein and individual foam positive proteins such as LPT, the level of bittering compounds from hops and wort color and would therefore give brewers more control.

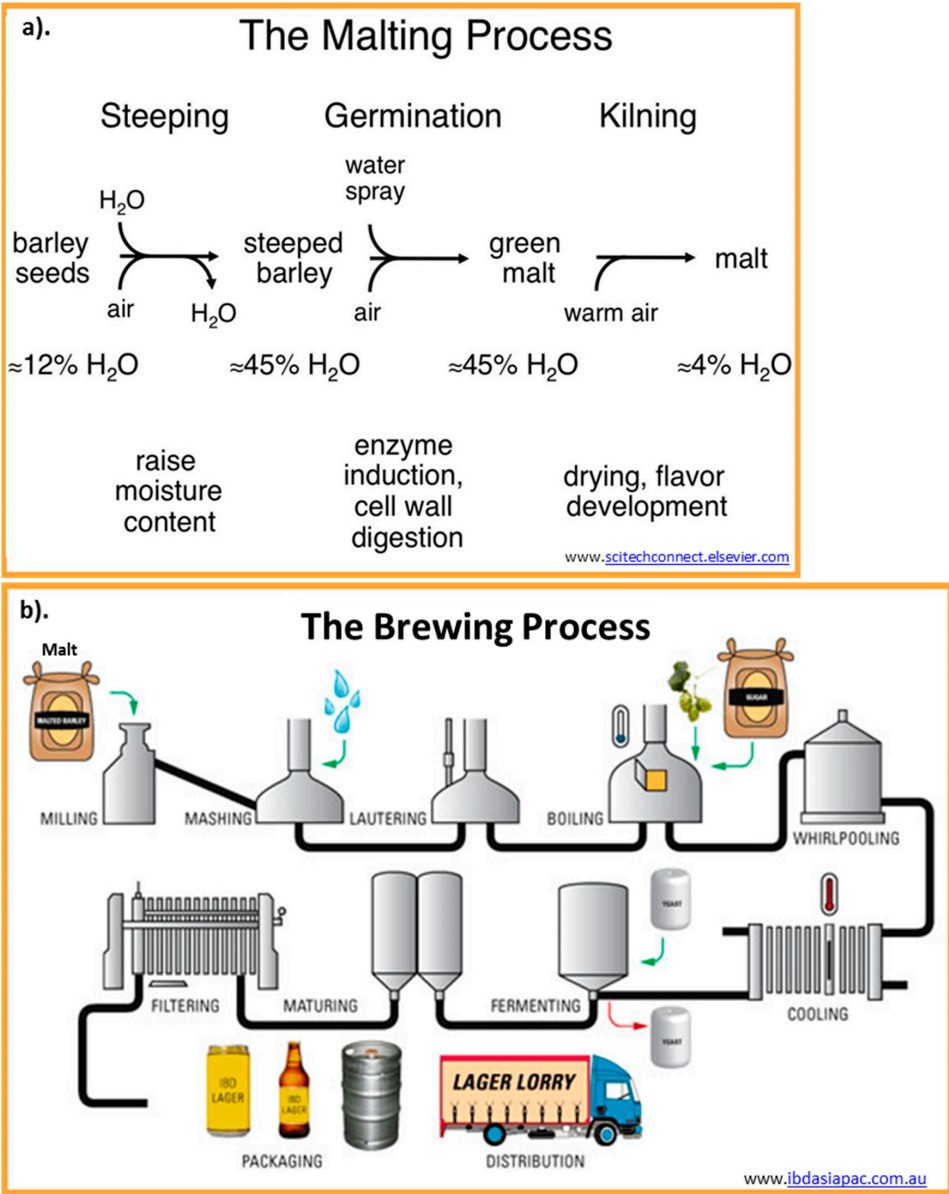

**Figure 2.** The stages of malting and brewing through to packaging [59]. Reproduced with permission from Gous P and Fox G, [Amylopectin synthesis and hydrolysis—understanding isoamylase and limit dextrinase in barley (Hordeum vulgare)]; published in Trends in Food Science and Technology (Elsevier), [2017].

The first test of wort quality is a measure of the total solubilized malt components, usually reported as hot water extract (HWE (%)) or as degrees Plato (°P). Extract is the main specification on a malt analysis sheet and it is used to calculate brewhouse yield. There are numerous reports for using NIR or MIR to predict malt extract in malt from a standard laboratory mash [6,7,60–63] but other reports using industry style mash show NIR and MIR can also be calibrated for wort Plato or Brix [64–67] which are more common brewing nomenclature. Through it has been shown, using partial least squares—discriminant analysis (PLS-DA), it was possible to predict lautering efficiency [68]. Clearly, all these studies, using different malts, mashing conditions and infrared instrumentation show the potential of measuring the total amount of solubilized components in wort. There will be many brewers who would use the real-time wort content to make any adjustments in the mash, the additional benefit is the use of the data for future mashing control.

Within the post-mash wort, and prior to boiling where hop compounds will add bitterness, there are hundreds of compounds from the malt [28]. Two key groups of compounds are required for the fermentation process, with these being sugars and amino acids. Sugars are more easily measured with high performance liquid chromatography or gas chromatography, although the former is the easier of those two. Maltose (glucose–glucose) is the dominant fermentable sugar and beta-amylase specifically cleaves maltose from amylose and amylopectin chains. Maltotriose (glucose–glucose–glucose) and glucose are the other remaining fermentable sugars, whereas the longer sugars such as maltotetrose are not fermentable. Near infrared and MIR have been used for calibrations for fermentable sugars [63,67,69,70]. Along with the capacity to measure the fermentable sugars, reports have shown NIR can measure fermentation efficiency or what is called the apparent attenuation limit (AAL) [71]. In addition, there have been a number of reports showing NIR and MIR calibrations specifically for ethanol [57,70,72,73] with specific studies reporting a flow-through FTIR for ethanol measurement [57,72]. The latter would be ideal during and post-fermentation to ensure targeted ethanol levels were achieved.

The second important group of compounds solubilized during mashing are proteins. As mentioned above, there are hundreds of individual proteins and peptides that survive the kilning process of malting and further heating steps of mashing and boiling during brewing [26,27]. The amount of total protein solubilized during mashing is called soluble protein. The soluble protein is also used in a ratio calculation to total malt protein and reported as protein modification or frequently called Kolbach Index (KI) in the malt quality specification. A higher soluble protein would indicate a high modification of protein during malting and thus more proteins will be solubilized in the mash. The measure of the soluble protein from the mash does not provide information on active proteins but the measure is based on the amount of nitrogenous compounds (mostly proteins). NIR has been used to build a calibration to predict soluble protein calibration [7,37,45,63,71]. These calibrations were developed with different malt samples and different mashing programs thus showing the robustness of NIR and MIR for samples with high concentrations of nitrogenous material (>4%).

Separate to the soluble protein measurement is a test for the total amount of free amino nitrogen (FAN) or an indication of amino acids available for yeast. Most brewers will require the FAN to be above a certain amount which indicates the yeast will have sufficient amino acids. NIR and MIR has been shown to also be suitable for calibrations for FAN [7,63,69,71] and more importantly for individual amino acids [37] with yeast having a preference for certain amino acids. The combination of soluble protein, FAN and amino acid profile would provide brewers with a vast amount of data for decisions on optimizing yeast pitching. Alternatively, if the brewery is locked into preprogramed brewing then any soluble protein (SP), FAN and amino acid data collected would be useful in a quality performance system or monitoring consistency of the brewing process.

Another major malt component is the non-starch polysaccharides, beta-glucan and arabinoxylan. These two polymers, negatively impact on wort by increasing viscosity which causes problems during filtration of wort and beer [49]. Wort beta-glucan is a routine measurement for malt quality for the sale of malt but it is not measured during commercial mashing where it should be tested after

mashing, however it has been reported that FTIR was unable to be calibrated for wort viscosity [61], a measured of both beta-glucan and arabinoxylan. However, this still presents an opportunity to develop calibrations for wort beta-glucan and arabinoxylan for in-line monitoring to ensure the wort and fermented beer with appropriate levels of beta-glucan and arabinoxylan.

Other areas where NIR has been issued in the brewing process is to assess bitterness [57,66,74–76], wort and beer color [57], pH [57], biomass (where the authors referred to a change in the optical density during fermentation) [65] and organic acids such as citric acids [66].

### 1.4.2. Fermentation

One of the final stages of brewing is fermentation (Figure 2), where the sugars from malt and solubilized in the wort will be consumed by yeast to produce alcohol. Several studies have shown the potential to assess the fermentability of wort using NIR [63,77,78]. This would be of great benefit to the brewers, as with the data of wort sugars, brewers could confidentially pitch yeast to obtain an optimal fermentation, rather than risking fermentation finishing too soon or too late which is a cost. In addition, having that information could help them pitch the yeast at a rate which is optimal and avoid losses when excess yeast being pitched and there is too much beer removed with that yeast.

### 1.4.3. Post-Production

Beer is a product that will spoil either from internal beer spoilage compounds such as excess lipid or lipoxygenase activity, but also through heat or allowing oxygen to be absorbed into the packaged products. Another way is the presence of spoilage microbes which create off-flavors. However, FTIR has been shown to detect spoilage bacteria, *Lactobacilli* and *Pediococci* as well as *Megasphaera* and *Pectinatu* where specific kits were used for the first two organisms combined and the latter two organisms combined. While FTIR calibrations were not to the authors expectations, it could provide an early warning of possible losses from spoilage bacteria [79].

## 2. Conclusions

All of these applications described here show the potential of predicting numerous wort and beer characteristics and when multiple traits can be predicted simultaneously the brewer will have more control of the process, or at least be able to use the data to trouble shoot when processing goes out. A number of in-line probes or optical sensors could be added during the brewing process, including mashing, lautering, boiling and fermentation. Currently, brewers may take a subsample during these stages to measure a parameter such as °Plato after the mash. This subsample could be tested using an infrared device and provide °Plato but also sugars and amino acid profiles. This information would provide brewers with data more useful in the next stages, especially fermentation. The challenge is now for instrument companies to work with brewers with advice and support on the best hardware options and opportunities for trait calibrations for more relevant data, resulting in more efficient brewing.

**Funding:** This research received no external funding.

**Acknowledgments:** The author acknowledges the University of California and the University of Queensland for the support in providing time to write this review.

**Conflicts of Interest:** The author declares no conflict of interest.

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
