# Peer review of "The Brewing Industry and the Opportunities for Real-Time Quality Analysis Using Infrared Spectroscopy"

_applsci, doi:10.3390/app10020616_

Round 1
Reviewer 1 Report
The author presented study about the opportunities for real time quality analysis using infrared in brewing industry. The subject of the work corresponds to scope of the Applied Science journal, the style and design of the work is very good and it is pleasant to read.
However, before publication the authors should revise the points listed below:
1) Line 176-180: It would be enough to add this citation [36] at the end of paragraph, i.e. on line 180
2) Line 199: Full citation of Figure 1 is missing.
3) Line 270-271: This is very interesting. Please could you describe this topic a little broader? How and which spoilage bacteria characteristic for beer spoilage could be detectable by FTIR analysis? This is an important factor affecting product quality and should be discussed further.
4) Line 272-272: Please adjust the line spacing to the rest of the manuscript
Author Response
The author presented study about the opportunities for real time quality analysis using infrared in brewing industry. The subject of the work corresponds to scope of the Applied Science journal, the style and design of the work is very good and it is pleasant to read.
Thank you
However, before publication the authors should revise the points listed below:
1) Line 176-180: It would be enough to add this citation [36] at the end of paragraph, i.e. on line 180.
This was corrected and the citation was deleted from the middle of the sentence.
2) Line 199: Full citation of Figure 1 is missing.
A full citation has been added for what is now Figure 2. I added another Figure which meets the comments from Reviewer 2.
3) Line 270-271: This is very interesting. Please could you describe this topic a little broader? How and which spoilage bacteria characteristic for beer spoilage could be detectable by FTIR analysis? This is an important factor affecting product quality and should be discussed further.
Thank you for your comments. Additional text has been added to include the bacterial strains and reference method.
4) Line 272-272: Please adjust the line spacing to the rest of the manuscript
Line spacing has been corrected.
Reviewer 2 Report
Review
Manuscript ID: applsci-656422
Title: The Brewing Industry and the Opportunities for Real Time Quality Analysis Using
Infrared
Brief summary/generic comments:
The author describes in this review the application of real time analysis of raw material, wort, and beer constituents using infrared spectroscopy or spectrometry. The manuscript gives a good overview of possibilities where infrared spectrometry can be applied within the brewing industry; though, detailed information is lacking. The manuscript would benefit from a section where the general principle of NIR/MIR, etc. is briefly explained and where the physical/chemical backgrounds (absorption of functional groups, etc.) are commented on. Furthermore, the syntax and spelling should be corrected as there are numerous mistakes; e.g. “line 23-24: “fermented beverages were [word missing] many thousands …”. Also, abbreviations should be first spelled out, and then abbreviated, such as e.g. LTP, PLS, ANN, etc.
Further points to be revised:
Page 1, title: Change to: “The Brewing Industry and the Opportunities for Real Time Quality Analysis Using Infrared Spectroscopy”. Also, the manuscript text should be revised as the author often speaks of NIR and MIR without declaring if spectrometry or spectroscopy is to be used.
Page 1, line 15-17: Please rephrase the section starting with “Importantly …”. What is meant by “…measuring wort and beer composition rather than just content.”?
Page 1, line 18: Please remove “especially for craft brewers” as this information is redundant and also is not reasoned.
Page 2, lines 59-66: Please provide reference(s) for this section.
Page 2, lines 75-78: The sentence starting with “Thus […]” needs to be clarified to improve readability.
Page 3, line 84: References are missing.
Page 3, line 104: Alpha-Amylase is partly denatured during mashing, though, but is still shows activity during lautering. The sentence should therefore be rephrased.
Page 3, line 122: Please correct to “[…] a minimum of 150 to 180 mg/L to …”
Page 4, lines 152-153: Please rephrase sentence starting with “Both of these …”.
Page 182, line 182: NIR is not used to determine bitterness in hops but to determine the amount of hop humulones or alpha-acids. Also, in beer, bitterness cannot be quantitated by NIR/FTIR but the concentration of bitter tasting substances such as iso-humulones can be determined.
2
Page 4, line 188: I also like beer to be phrased as “liquid gold” but still, please rephrase the header.
Page 6, lines 227-231: In this section, the author is addressing to mashing. Yet, the Kolbach Index is a malt quality parameter. Even though the information given here are not wrong, it can still be misleading to the reader to mention the Kolbach Index here. I would therefore recommend to move the paragraph about the Kolbach Index to the malt section, and/or to clarify this section.
Page 6, lines 262: Please provide references for the application of IR for biomass and organic acids.
Page 6, lines 263-266: “Bitterness” is not a flavor, but it is a gustatory perception (taste). Please clarify.
Even though Pale Ales and India Pale Ales are bitter, the hop character is derived from hop volatiles/hop essential oils which are not yet measured by IR spectroscopy. This paragraph is therefore misleading and should be rephrased to improve clarity for the reader.
Author Response
The author describes in this review the application of real time analysis of raw material, wort, and beer constituents using infrared spectroscopy or spectrometry. The manuscript gives a good overview of possibilities where infrared spectrometry can be applied within the brewing industry; though, detailed information is lacking. The manuscript would benefit from a section where the general principle of NIR/MIR, etc. is briefly explained and where the physical/chemical backgrounds (absorption of functional groups, etc.) are commented on.
An additional section has been added towards the start of the manuscript with the sub-heading Background on infrared (near and mid)
Furthermore, the syntax and spelling should be corrected as there are numerous mistakes; e.g. “line 23-24: “fermented beverages were [word missing] many thousands …”.
This has been rephrased
Also, abbreviations should be first spelled out, and then abbreviated, such as e.g. LTP, PLS, ANN, etc.
These acronyms have been named before being used in the document.
Further points to be revised:
Page 1, title: Change to: “The Brewing Industry and the Opportunities for Real Time Quality Analysis Using Infrared Spectroscopy”.
Title has been changed as per suggestion
Also, the manuscript text should be revised as the author often speaks of NIR and MIR without declaring if spectrometry or spectroscopy is to be used.
Thank for you this important point. Clarification is made in the first instance in the Abstract then through out the document.
Page 1, line 15-17: Please rephrase the section starting with “Importantly …”. What is meant by “…measuring wort and beer composition rather than just content.”?
As this is the abstract, I have added one sentence to save on word count. The new text has been added to clarify the meaning of ‘composition rather than just content’. “This includes measuring individual sugars and amino acids prior to fermentation, rather than total oPlato or Free Amino Acids content.”
Page 1, line 18: Please remove “especially for craft brewers” as this information is redundant and also is not reasoned.
These words have been deleted.
Page 2, lines 59-66: Please provide reference(s) for this section.
A number of references have been added to this section.
Page 2, lines 75-78: The sentence starting with “Thus […]” needs to be clarified to improve readability.
Page 3, line 84: References are missing.
References have been added.
Page 3, line 104: Alpha-Amylase is partly denatured during mashing, though, but is still shows activity during lautering. The sentence should therefore be rephrased.
This sentence has been rephrased with references added.
Page 3, line 122: Please correct to “[…] a minimum of 150 to 180 mg/L to …”
Corrected
Page 4, lines 152-153: Please rephrase sentence starting with “Both of these …”.
This sentence has been rephrased.
Page 182, line 182: NIR is not used to determine bitterness in hops but to determine the amount of hop humulones or alpha-acids. Also, in beer, bitterness cannot be quantitated by NIR/FTIR but the concentration of bitter tasting substances such as iso-humulones can be determined.
This has been rewritten with additional references to clarify.
2
I am unsure of this?
Page 4, line 188: I also like beer to be phrased as “liquid gold” but still, please rephrase the header.
Rephased
Page 6, lines 227-231: In this section, the author is addressing to mashing. Yet, the Kolbach Index is a malt quality parameter. Even though the information given here are not wrong, it can still be misleading to the reader to mention the Kolbach Index here. I would therefore recommend to move the paragraph about the Kolbach Index to the malt section, and/or to clarify this section.
This has been rephrased. The soluble protein is also used in a ratio calculation to total malt protein and reported as protein modification or frequently called Kolbach Index (KI) in the malt quality specification. A higher soluble protein would indicate a high modification of protein during malting and thus more proteins will be solubilized in the mash.
Page 6, lines 262: Please provide references for the application of IR for biomass and organic acids.
Additional references and information has been added.
Page 6, lines 263-266: “Bitterness” is not a flavor, but it is a gustatory perception (taste). Please clarify.Even though Pale Ales and India Pale Ales are bitter, the hop character is derived from hop volatiles/hop essential oils which are not yet measured by IR spectroscopy. This paragraph is therefore misleading and should be rephrased to improve clarity for the reader.
These statements have been deleted.
Reviewer 3 Report
The article concerns important topic regarding the use of different IR techniques through beer production process. The strengths of the manuscript include comprehensive description of important ingredients and brewing process itself. Unfortunately the area of the article related to NIR and MIR is very concise, and recalls a chapter in monograph or article in industry magazine rather than a review paper in scientific journal. The whole article should be reconsidered and rewritten including important NIR and chemometrics data, facilitating the reader and developing understanding of the limitations of applied methods. There is a lack of in-depth critical discussion regarding the opportunities for the industry, mentioned in the article title. References are not edited according to instructions. Moreover, I would like to suggest reorganization of manuscript sections, some are very short i.e. 1.2. Hops, whereas 1.3 Production of beer could be divided into several sections corresponding to individual stages of production.
Line 18 The statement regarding craft brewers is doubtful, lot of craft breweries around the world still can’t afford NIR equipment.
Line 198 The quality(resolution/compression parameters) of the figure is to low.
Author Response
The article concerns important topic regarding the use of different IR techniques through beer production process. The strengths of the manuscript include comprehensive description of important ingredients and brewing process itself. Unfortunately the area of the article related to NIR and MIR is very concise, and recalls a chapter in monograph or article in industry magazine rather than a review paper in scientific journal.
A separate section has been added to describe NIR and MIR. This is the same point noted by reviewer 2.
The whole article should be reconsidered and rewritten including important NIR and chemometrics data, facilitating the reader and developing understanding of the limitations of applied methods.
The manuscript has a number of additions and clarification of technical inconsistencies with additional referencing.
There is a lack of in-depth critical discussion regarding the opportunities for the industry, mentioned in the article title.
The Conclusion has been expanded to include opportunities for the industry.
References are not edited according to instructions.
References have been edited for style
Moreover, I would like to suggest reorganization of manuscript sections, some are very short i.e. 1.2. Hops, whereas 1.3 Production of beer could be divided into several sections corresponding to individual stages of production.
The hops section has been expanded. The section on production of beer has had subheadings added.
Line 18 The statement regarding craft brewers is doubtful, lot of craft breweries around the world still can’t afford NIR equipment.
This has been rephrased.
Line 198 The quality(resolution/compression parameters) of the figure is to low.
A number figure has been added which is now figure 2.
Round 2
Reviewer 3 Report
Thank you for revised version of your manuscript. I would like to suggest only few minor changes.
Please change the title back to “The Brewing Industry and…”, please include this change in the rest of the manuscript.
Inserted in the revised version, figure 1. is of poor quality (blurry text and colors of the bar).
Author Response
Thanks you for your comments. I have correct the title and throughout the text where an autocorrect changed some words.
For Figure 1, this is the best image I have available.